# Sex-Related Differences in Left Atrial Low-Voltage Areas According to CHA_2_DS_2_-VA Scores among Patients with Atrial Fibrillation

**DOI:** 10.3390/jcm11113111

**Published:** 2022-05-31

**Authors:** Do Young Kim, Yun Gi Kim, Ha Young Choi, Yun Young Choi, Ki Yung Boo, Kwang-No Lee, Seung-Young Roh, Jaemin Shim, Jong-Il Choi, Young-Hoon Kim

**Affiliations:** 1Division of Cardiology, Department of Internal Medicine, Korea University Medical Center, Seoul 02841, Korea; sneeze_kr@naver.com (D.Y.K.); tmod0176@gmail.com (Y.G.K.); yych60@naver.com (Y.Y.C.); rsy008@gmail.com (S.-Y.R.); jongilchoi@korea.ac.kr (J.-I.C.); yhkmd@unitel.co.kr (Y.-H.K.); 2Division of Cardiology, Dongtan Sacred Heart Hospital, Hallym University College of Medicine, Hwaseong 18450, Korea; 3Division of Cardiology, Soonchunhyang University Chonan Hospital, Chonan 31151, Korea; inter0019@naver.com; 4Division of Cardiology, Jeju National University Hospital, Jeju 63241, Korea; pidori@hanmail.net; 5Department of Cardiology, Ajou University School of Medicine, Suwon 16499, Korea; knlee81@naver.com

**Keywords:** atrial fibrillation, catheter ablation, female, sex, electro-anatomical remodeling, low voltage area

## Abstract

(1) Background: We hypothesized that female sex would have a differential impact on left atrial (LA) low-voltage areas (LVAs) according to CHA_2_DS_2_-VA scores. (2) Methods: This study included 553 patients who underwent radiofrequency catheter ablation (RFCA) for atrial fibrillation (AF). LVAs were defined as regions with bipolar peak-to-peak voltages of <0.5 mV. The proportion of LVAs was calculated by dividing the total LVA by the LA surface area. (3) Results: There was no sex-related difference in LA LVAs among patients with a CHA_2_DS_2_-VA scores ≤ 2. The proportion of LVAs was significantly higher in females among patients with CHA_2_DS_2_-VA scores of 3 or 4 (10.1 (4.7–15.1)% vs. 15.8 (9.2–32.1)%; *p* = 0.027). Female sex was significantly associated with extensive LVAs (LVA proportion ≥ 30%). Females had odd ratios of 27.82 (95% confidence interval (CI) 3.33–756.8, *p* = 0.01), and 1.53 (95% CI 0.81–2.83, *p* = 0.184) for extensive LAVs in patients with CHA_2_DS_2_-VA scores ≥ 3 and CHA_2_DS_2_-VA scores < 3, respectively. In the multiple regression model, female patients with a CHA_2_DS_2_-VA ≥3 were significantly associated with a higher proportion of LVAs (β = 8.52, *p* = 0.039). (4) Conclusions: Female sex was significantly associated with extensive LVAs, particularly when their CHA_2_DS_2_-VA scores were ≥3. This result suggests that female sex has a differential effect on the extent of LVAs based on the presence of additional risk factors.

## 1. Introduction

Patients with atrial fibrillation (AF) are at a higher risk of ischemic stroke, and the risk varies depending on the presence of risk factors [1]. Despite the higher incidence of AF in males [2], numerous clinical studies have demonstrated that the risk of ischemic stroke is greater in females than in males with AF [3,4,5,6,7,8]. This has resulted in the inclusion of female sex into stroke risk stratification models, including the CHA_2_DS_2_-VASc scoring system [9,10]. However, there is still debate over the role of female sex in stroke risk stratification among patients with AF [11,12]. Recent studies have suggested that compared to male counterparts, the contribution of female sex on stroke risk is dependent upon other risk factors, and female sex does not confer excess stroke risk in patients without additional risk factors [1,7,11,13]. Conversely, it has been found that females who are older [11] or have additional stroke risk factors are at higher risk of stroke [13,14]. Hence, it is suggested that female sex acts as a stroke risk modifier, rather than as a risk factor [13]. However, the underlying mechanism explaining this differential effect of female sex on stroke risk is poorly understood [12,15]. Electro-anatomical atrial remodeling involving atrial fibrosis is the mainstay in the pathogenesis of AF, and atrial fibrosis contributes to the risk of stroke in these patients [16]. Left atrial (LA) fibrosis can be quantified as bipolar low-voltage areas (LVAs) using a 3D electro-anatomical mapping system, and the extent of LVAs is correlated with the presence of ischemic stroke [17]. Based on these studies, we hypothesized that the differential effect of female sex on stroke risk is associated with the extent of LVAs. Therefore, we sought to investigate the differential impact of female sex on electro-anatomical atrial remodeling according to the presence of additional risk factors. This was achieved by comparing the extent of LVAs between male and female subjects and correlating these to their CHA_2_DS_2_-VA risk scores.

## 2. Materials and Methods

### 2.1. Cohort Creation

The radiofrequency catheter ablation (RFCA) registry of the Korea University Medical Center Anam Hospital was utilized [18]. We retrospectively reviewed patients who underwent RFCA for AF between January 2010 and December 2018 using a 3D mapping system (NavX System, Abbott., St. Paul, MN, USA). Among them, we included patients with 3D electro-anatomical voltage mapping that was performed during sinus rhythm and was completed prior to RFCA. We excluded patients with electro-anatomical voltage mapping with insufficient mapping points (<800 points) and insufficient coverage of the LA (Appendix A). Finally, we reviewed 553 patients, and all the subjects were adult Asian patients. Electrograms with amplitudes >0.5 mV were defined as normal potentials, and low-voltage areas were defined as contiguous areas with a bipolar voltage ≤0.5 mV [19]. The proportion of LVAs was calculated by dividing the total LVA by the LA surface area. This study was approved by the institutional review board of the Korea University Anam Hospital.

### 2.2. Electro-Anatomical Mapping and Ablation Procedure

Electro-anatomical mapping was performed during sinus rhythm prior to RFCA. The ablation procedure protocol has been previously summarized [18,20]. Amiodarone was interrupted at least 4 weeks before RFCA, and other antiarrhythmic medications were discontinued at least 1 week prior to the procedure. Circumferential antral pulmonary vein (PV) isolation was performed in all patients. Additional ablations, such as complex fractionated atrial electrogram-guided ablation or linear ablation, were performed at the operator’s discretion if sustained AF was induced after PV isolation. The endpoint of the procedure was the elimination of non-PV triggers and non-inducibility of sustained atrial arrhythmia for paroxysmal AF and non-paroxysmal AF, respectively. Electro-anatomical voltage maps of the left atrium, excluding the pulmonary veins, were created using a 20 pole circular mapping catheter (AFocus catheter, Abbott., St. Paul, MN, USA).

### 2.3. Follow-Up

Patients visited the outpatient clinic at 1, 3, 6, 9, and 12 months after RFCA and every 6 months thereafter. The follow-up protocol included regular 12 lead electrocardiography at each visit and Holter monitoring at 3, 6, 9, and 12 months after RFCA. Patients were encouraged to contact or visit the outpatient clinic if they experienced possible symptoms of AF recurrence. Patients were prescribed anticoagulation therapy for 2–3 months after RFCA. Patients received antiarrhythmic drugs during the 3-month blanking period.

### 2.4. Statistical Analysis

Continuous variables were expressed as means and standard deviations, and categorical data were expressed as numbers and percentages. Non-parametrically distributed data were reported as median values with interquartile ranges. For comparisons across groups, continuous variables were compared using Student’s t-test, and categorical variables were analyzed using the chi-square test or Fisher’s exact test, as appropriate. Multiple linear regression analysis was performed to evaluate the factors influencing the low-voltage area. Patient ages, CHA_2_DS_2_-VA scores ≥ 3, types of AF, AF durations, LA diameters ≥ 45 mm, LV ejection fractions, and estimated glomerular filtration rates (eGFR) were entered into the model based on a previous study [21]. Stepwise backward elimination and all subsets’ regression were performed to select the final linear regression model. Multiple logistic regression analysis was used to investigate the independent factors that predict the presence of extensive LVA. The extent of LVA was categorized into stage I (<5%), II (≥5–<20%), III (≥20%–<30%), and IV (≥30%) [22]. Accordingly, the extensive LVA burden was defined as stage IV (LVA proportion ≥30%). Differences were considered statistically significant at *p* < 0.05. Statistical analyses were performed using R (version 3.2.1; R Foundation for Statistical Computing, Vienna, Austria).

## 3. Results

### 3.1. Baseline Demographics

Baseline characteristics are presented in Table 1. A total of 553 patients with AF who underwent RFCA for the first time were included in this study. The mean age of the whole population was 55.9 ± 10.6 years, and 109 patients (19.7%) were female. The female patients were significantly older than the male patients and were less likely to have non-paroxysmal AF. No significant differences were found between the two sex groups in terms of CHA_2_DS_2_-VA score, duration of AF, LA diameter, indexed LA volume, proportion of LA LVAs, and left ventricular ejection fraction (LVEF).

### 3.2. LA LVAs According to Number of Non-Sex Category Risk Factors (CHA_2_DS_2_-VA Scores)

The proportion of LA LVAs according to CHA_2_DS_2_-VA scores is presented in Appendix A. The median of LA LVAs proportion was the lowest in the group with CHA_2_DS_2-_VA scores of 0 and 2. It was the highest in the group with CHA_2_DS_2_-VA scores of 5. However, LA LVA proportion changes according to CHA_2_DS_2_-VA scores were not a statistically significant trend.

### 3.3. Sex-Related Difference of Proportion of LVAs According to CHA_2_DS_2_-VA Scores

Figure 1 shows the left atrial bipolar voltage maps of representative female and male patients. There was no significant difference in the proportion of LVA between males and females among patients with a CHA_2_DS_2_-VA score of 0 (9.3 (4.9–15.9) vs. 8.1 (4.8–15.8) *p* = 0.957), 1 (10.0 (4.7–15.0) vs. 9.4 (5.8–17.8), *p* = 0.536), and 2 (8.6 (5.8–14.8) vs. 9.6 (6.5–26.9), *p* = 0.398). There were no female counterparts among patients with a CHA_2_DS_2_-VA score of 5. Females with CHA_2_DS_2_-VA scores of 3 or 4 had a significantly higher proportion of LVAs than males (10.1 (4.7–15.1) vs. 15.8 (9.2–32.1), *p* = 0.027; Figure 2; Appendix A).

### 3.4. Differential Effect of Female Sex on the Extent of LVAs

Appendix A shows baseline characteristics according to the four LVA stages. The proportion of females, prevalence of LA diameter ≥ 45 mm and non-paroxysmal AF, and eGFR were significantly different across the four groups. Patients with the most extensive LVA burden (stage IV) had the highest proportion of females, the highest prevalence of LA diameter ≥ 45 mm and non-paroxysmal AF, and the longest duration of AF. Multiple linear regression analysis and logistic regression analysis were performed to determine the differential effect of female sex according to CHA_2_DS_2_-VA scores. In the multiple logistic model adjusted for age, duration of AF, type of AF, LA diameter, LVEF, non-sex criteria risk factors, and eGFR, female patients were significantly associated with extensive LVA burden (LVAs proportion of ≥30% (stage IV)) in the entire study population (odd ratio (OR) 1.93, 95% confidence interval (CI) 1.07–3.45, *p* = 0.027). In the subgroup analysis, female sex was an independent predictor of extensive LVA extension among patients with a CHA_2_DS_2_-VA score ≥3 (OR 27.82, 95% CI 3.33–754.8, *p* = 0.01). However, female sex was not significantly associated with extensive LVA burden among patients with a CHA_2_DS_2_-VA score <3 (OR 1.54, 95% CI 0.81–2.83, *p* = 0.184). Female sex remained a significant predictor of extensive LVA burden in patients with hypertension (HTN) or stroke (Figure 3).

In the multiple linear regression model, female sex with a CHA_2_DS_2_-VA score ≥3 (β = 8.52, *p* = 0.039) remained an independent factor associated with higher low-voltage area proportions after adjusting for age, duration of AF, type of AF, LVEF, eGFR, and LA diameter (Table 2). In this regression model, non-paroxysmal AF, AF duration, and LA diameter ≥ 45 mm were identified as the independent predictors for extensive LVA burden.

## 4. Discussion

In this study, sex-related differences in LA fibrosis in patients with AF were examined by comparing low-voltage areas. LVAs were significantly higher in females than in males among patients with CHA_2_DS_2_-VA scores of 3 or 4. However, the difference was not significant among patients with CHA_2_DS_2_-VA scores <3. Female sex was associated with the presence of extensive LVA burden, which was defined as an LVA proportion ≥30%; however, the association was excessively evident among females with CHA_2_DS_2_-VA scores ≥3. Female sex with a CHA_2_DS_2_-VA score ≥3 remained a significant factor that was correlated with LVA proportion even after adjusting for LA diameter, GFR, type of AF, duration of AF, non-sex criteria risk factors, and age.

### 4.1. Left Atrial Fibrosis in Patients with AF

AF and atrial remodeling are closely intertwined, and structural remodeling of the atrium is characterized by atrial enlargement and atrial fibrosis [16]. Modern cardiac imaging methods can quantify the degree of atrial structural remodeling, and its measurements have been reported to be correlated with the outcome of AF [16,18,23,24]. Among cardiac imaging methods, atrial fibrosis can be quantified by delayed enhancement in the cardiac MRI and LVA of the electro-anatomical 3D mapping system [16,17]. Atrial fibrosis is essential in the pathophysiology of AF. Atrial fibrosis provides a substrate for atrial fibrillation by causing atrial local conduction abnormalities [16,25].

### 4.2. Sex-Related Difference in the Extent of LA Fibrosis among Patients with AF

Prior studies have reported that female patients with AF have a higher burden of atrial fibrosis [21,26,27,28], and female sex was an independent risk factor for a higher burden of atrial fibrosis even after adjusting for biomarkers, type of AF, age, time from AF diagnosis, and LA volume [13,21]. In a study that analyzed LA tissues obtained from patients with AF, the sex-related difference in LA fibrosis were related to the differential expression of fibrosis-related genes and proteins [29]. In this study, female sex was an independent predictor of a higher extent of LVA, and this result was consistent with previous results. Additionally, our study highlighted that the association between females and LA fibrosis was more evident among patients with three or more than three risk factors. This result suggests that female sex has a differential effect on the extent of LA fibrosis according to the presence of additional risk factors. A possible explanation for the differential effect of females on LA fibrosis is menopause-related estrogen deficiency. Menopause-related circulating estrogen level changes may contribute to increases in the cardiac extracellular matrix, which is a key element of atrial fibrosis [30,31].

### 4.3. Female Sex as a Risk Modifier on Stroke Risk and LA Fibrosis

While the risk of stroke in females without other risk factors is comparable to that of their male counterparts, the risk increases in females with additional risk factors other than sex [12,15]. The impact of female sex on the risk of stroke is intricately linked to the presence of other risk factors, and female sex acts as a risk modifier for stroke risk in patients with AF, not a risk factor [13]. However, the exact mechanism of the differential effect of female sex on stroke risk in AF patients has not been clearly elucidated. Previous studies have shown that the degree of left atrial fibrosis assessed by delayed enhancement MRI or LVA is correlated with the risk of ischemic stroke [17,23,32]. The results of our study suggest that the impact of female sex on LA fibrosis is dependent on the presence of additional risk factors. Taken together, it is conceivable that the differential effect of female sex on stroke risk is strongly associated with LA fibrosis.

### 4.4. Limitations

This study had several limitations. First, this was a retrospective observational study, and this study enrolled a limited number of patients with CHA_2_DS_2_-VA scores ≥ 3. Therefore, the results of our study need to be validated in a prospective cohort study with a large number of subjects. Second, the patients’ menopausal status, which may be an important factor affecting the differential effect of female sex on LA fibrosis, was not evaluated. Third, patients with ongoing atrial fibrillation underwent electrical cardioversion prior to the voltage mapping. Atrial stunning has been reported after cardioversion [33]. Although the effect of atrial stunning on voltage mapping has not been investigated, voltage mapping following cardioversion might be affected by atrial stunning. Fourth, LA diameter was used to estimate left atrial volume assessment. Given that left atrial volume index provide more accurate LA volume assessment, left atrial diameter could be a poor indicator of actual LA remodeling. Finally, a newer version of the high-density mapping catheter was not used for voltage mapping. However, we tried to maintain the quality of the voltage mapping by excluding patients who have voltage maps with voltage points less than 800 points.

## 5. Conclusions

Female sex was an independent predictor of extensive LA LVAs. This association was more evident among female patients with three or more additional risk factors. These findings provide evidence for the differential effect of female sex on LA remodeling according to the presence of additional risk factors.

## Figures and Tables

**Figure 1 jcm-11-03111-f001:**
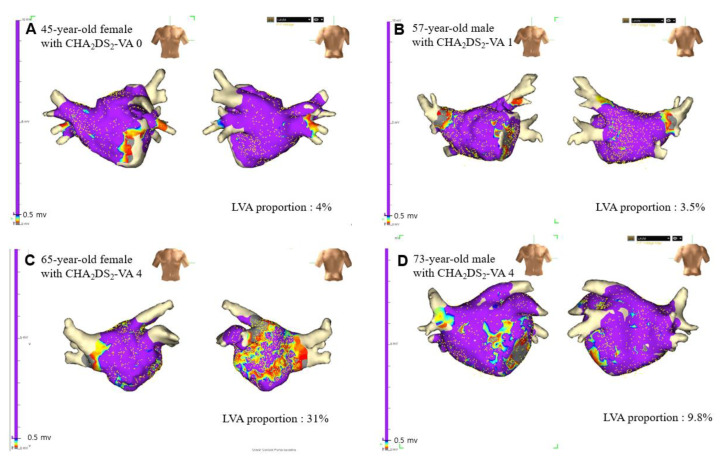
The left atrial bipolar voltage map of representative female (**A**,**C**) and male (**B**,**D**) patients. The color gradient indicates serial changes in the electrogram amplitude from purple at >0.5 mV to gray at <0.1 m. The proportion of low voltage area was 4% in a 45-year-old female patient with CHA_2_DS_2_-VA score of 0 (**A**) and 3.5% in a 57-year-old male patient with CHA_2_DS_2_-VA score of 1 (**B**). The burden of low voltage area was extensive (31%) in a 65-year-old female patient with CHA_2_DS_2_-VA score of 4 (**C**), while the low voltage area proportion was 9.8% in a 73-year-old male patient whose CHA_2_DS_2_-VA score was 4 (**D**).

**Figure 2 jcm-11-03111-f002:**
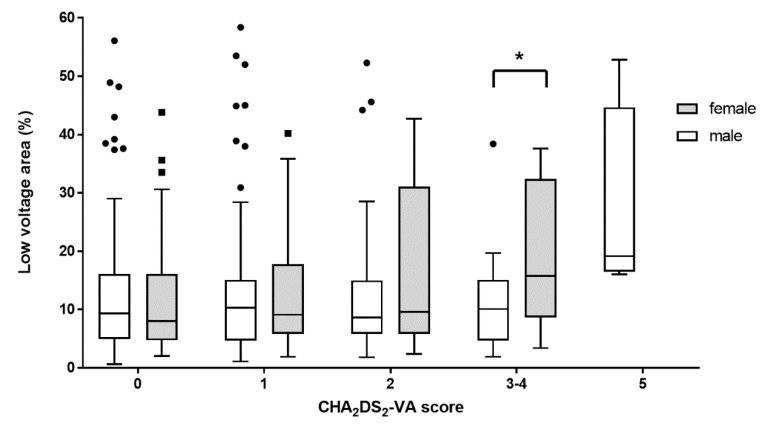
Difference in the proportion of left atrial low voltage area between males and females according to CHA_2_DS_2_-VA score. The proportion of left atrial low voltage area was similar between males and females in the patients with CHA_2_DS_2_-VA scores of 0 to 2. There was a significant difference in the proportion of low voltage area (LVA) between male and female in the patients with CHA_2_DS_2_-VA scores of 3 or 4 (10.1 (4.7–15.1)% vs. 15.8 (9.2–32.1)%; *p* = 0.027). Within each box, horizontal lines denote the median value; boxes extend from the first quartile to the third quartile value of each group. The vertical extended lines denote the adjacent values. The dots denote values outside the range of the adjacent values. * *p* = 0.027.

**Figure 3 jcm-11-03111-f003:**
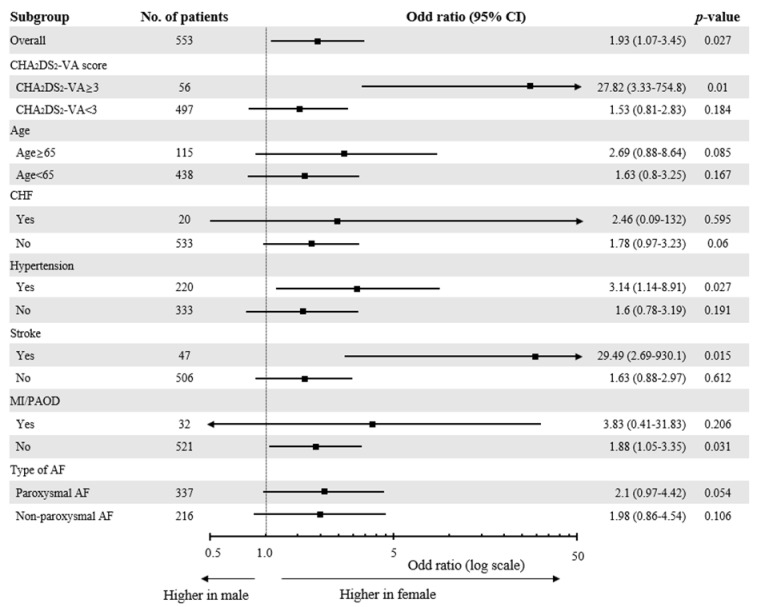
Multivariate logistic regression analysis performed to evaluate the association between female sex and extensive LVA burden (LVA proportion ≥ 30% (stage IV)). Age, number of non-sex criteria risk factors (heart failure, hypertension, diabetes mellitus, myocardial infarction, or peripheral artery disease), type of AF, AF duration, LA diameter ≥ 4.5 cm, LV ejection fraction, and estimated glomerular filtration rates (eGFR) were entered into the model. CHF, congestive heart failure; MI, myocardial infarction; PAOD, peripheral artery occlusive disease; AF, atrial fibrillation.

**Table 1 jcm-11-03111-t001:** Baseline characteristics.

	Male	Female	Total	*p*-Value
	(*n* = 444)	(*n* = 109)	(*n* = 553)	
Age, years	54.9 ± 10.6	59.9 ± 9.6	55.9 ± 10.6	**<0.001**
BMI, kg/m^2^	25.2 ± 3.0	24.5 ± 3.2	25.1 ± 3.1	**0.022**
CHF, *n* (%)	17 (3.8)	3 (2.8)	20 (3.6)	0.8
HTN, *n* (%)	174 (39.1)	46 (42.2)	220 (39.8)	0.641
DM, *n* (%)	29 (6.5)	8 (7.3)	37 (6.7)	0.929
MI/PAOD, *n* (%)	26 (5.9)	6 (5.5)	32 (5.8)	1
CHA_2_DS_2_-VA, *n* (%)				0.176
0	200 (45.1)	44 (40.4)	244 (44.1)	
1	126 (28.3)	30 (27.5)	156 (28.2)	
2	78 (17.6)	19 (17.4)	97 (17.5)	
3	32 (7.2)	12 (11.0)	44 (8.0)	
4	4 (0.9)	4 (3.7)	8 (1.4)	
5	4 (0.9)	0 (0.0)	4 (0.7)	
eGFR, ml/min per 1.73 m^2^	78.1 ± 13.9	73.7 ± 18.2	77.3 ± 15.0	**0.019**
Non-paroxysmal AF, *n* (%)	187 (42.1)	29 (26.6)	216 (39.1)	**0.004**
Duration of AF, years	3.0 (1.0–6.0)	3.0 (1.0–7.5)	3.0 (1.0–6.0)	0.96
LA diameter, mm	41.0 (37.5–44.5)	40.2 (36.0–42.8)	40.7 (37.4–44.3)	0.056
LVEF, %	57.5 (54.5–57.5)	57.5 (54.5–57.5)	57.5 (54.5–57.5)	0.249
LA low-voltage area (%)	9.6 (5.2–15.2)	9.8 (5.8–20.9)	9.7 (5.4–16.4)	0.152

Values are shown as means ± standard deviations, medians (interquartile ranges), and numbers (percentages) for continuous and categorical variables, respectively. The differences between the groups are presented as overall *p*-values. *p*-values marked with bold indicate statistically significant *p*-values. BMI, body mass index; CHF, congestive heart failure; HTN, hypertension; DM, diabetes mellitus; MI, myocardial infarction; PAOD, peripheral artery occlusive disease; eGFR, estimated glomerular filtration rate; AF, atrial fibrillation; LVEF, left ventricular ejection fraction; LA, left atrial.

**Table 2 jcm-11-03111-t002:** Univariate and multivariate associations of clinical variables with LA low voltage areas using linear regression model.

	Univariate Analysis			Multiple Analysis Model		
	Beta	95% CI	*p*-Value	Beta	95% CI	*p*-Value
Intercept				7.95	5.96–9.95	<0.001
Age per 10 years	1.37	−0.03–2.76	0.054			
Female with CHA_2_DS_2_-VA ≥ 3	6.46	−2.33–15.25	0.149	8.52	0.44–16.6	0.039
Non-PAF	12.93	10.09–15.77	<0.001	10.88	7.90–13.86	<0.001
AF duration, years	0.27	0.07–0.47	0.009	0.24	0.05–0.43	0.012
LA diameter ≥ 45 mm	11.37	7.89–14.85	<0.001	6.96	3.43–10.49	<0.001
LVEF, %	−0.33	−0.59–−0.07	0.014			
eGFR < 60 mL/min per 1.73 m^2^	6.13	1.14–11.12	0.016			

Adjusted R^2^ for the selected multiple lineal regression model = 0.166, CI, confidence interval; PAF, paroxysmal AF; LA, left atrial; LVEF, left ventricular ejection fraction; eGFR, estimated glomerular filtration rate.

## Data Availability

The data, analytical methods, and study materials will not be made available to other researchers for purposes of reproducing the results or replicating the procedure.

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
