# Peer review of "Sex-Related Differences in Left Atrial Low-Voltage Areas According to CHA2DS2-VA Scores among Patients with Atrial Fibrillation"

_jcm, 2022, doi:10.3390/jcm11113111_

Round 1
Reviewer 1 Report
Very well written paper about sex differences in left atrial LVAs.
The study is retrospective and there is some bias as patients were excuded if not enough mapped points, etc was present.
But all in all the results are novel.
One major mistake:
According to Figure 2. there are no females in the study with a CHADSVASC score with 5 and above. So the only scientific statement we can make is to compare CHADSVASC 3-4 pateints which shows significant sex differences. It is also not clear why CHADSVASC group 0 , 1 and 2 are presented separately and 3-4 combined. I guess because this shows statistical significance.
Please rewrite the text: write CHADSVASC 3-4 instead of CHADSVASC ≥3. This is what your data presents.
I would delete the paragraph : 4.1. Left atrial fibrosis in patients with AF
Since it has nothing to do with the present study.
Author Response
We thank the editor and the reviewers for their helpful and insightful comments. We believe that our revised manuscript, based on their comments and suggestions, has been markedly improved relative to its accuracy, readability, and the soundness of our conclusions. Please see the attachment.

Reviewer 2 Report
General comment:
The study data showed many meaningful data for the association between stroke risk and low voltage areas (LVAs) in males and females. The study demonstrated that LVAs were significantly higher in females than males in patients with CHA2DS2-VA scores of 3 or 4. Importantly, female sex with CHA2DS2-VA scores of 3 or 4 was associated with extensive LVA burden (≥30%). However, it is unclear that the definition of extensive LVA burden is ≥ 30%. How to define the extensive LVA? Is it an arbitrary number? Or, are there related studies for the definition of extensive LVA? Please clarify it in line 105. Also, are there any data related to each LVA burden (0-10%, 10-20%, ≥30%)? Based on each LVA burden, the data could be CHA2DS2-VA scores and the patient numbers, similar to Table S2.
Specific comment:
1) Need to clarify symbols (*, +) for the first and correspondence authors in line 15.
2) Need to change “Abott” to “Abbott” in lines 62 and 83.
3) Recommend to bold or highlight significant values (p < 0.05) in Table 1.
4) Need to change “was lowest in the group” to “was the lowest in the group” in line 125.
5) “The proportion of LA LVAs was the lowest in the group with CHA2DS2-VA scores of 0” in lines 125 and 126. However, in Table S1, CHA2DS2-VA scores of 0 and 2 are the same as the average value of 9.1. Please clarify it.
6) No mention of “Figure 2 and Table S2” in the 3.3 section (lines 129 – 132).
7) No mention of Figure 1 in the manuscript. Figure quality was bad (especially the voltage color scale bar). Also, it may need to add sex, LAVs number, and CHA2DS2-VA score into each figure.
7) Need to put a period(.) in line 135.
8) Need to change “male and female” to “males and females” in lines 141 and 142.
9) Need to change “for 0 to 2” to “of 0 to 2” in line 143.
10) Need to change “extend” to “extent” in line 148.
11) Need to explain more in detail about Table 2 in lines 160-162.
Author Response

(The authors gave the same response as above.)

Reviewer 3 Report
Dear Sir/Madam,
I had the opportunity to act as a reviewer on the recent submission by Kim et al. to the Journal of Clinical Medicine.
The authors present interesting original research regarding sex-related differences in left atrial low-voltage areas among patients with atrial fibrillation. They have retrospectively included a total of 553 patients undergoing catheter ablation of atrial fibrillation. The authors found that female sex was an independent predictor of high left atrial low voltage area burden.
The manuscript is well written and the results are interesting and of high clinical interest.
However, some major issues need to be addressed:
- According to the authors there have been included patients with 3D electro-anatomical voltage mapping that was performed during sinus rhythm and was completed prior to RFCA (lines 62-63). This raises a very important question, as many patients had non-paroxysmal AF: did the operators cardiovert the patients before starting the mapping? Stunning could be a significant confounder.
- I strongly recommend eliminating the concept of left atrial diameter, since this is measured only in one axis and indicating the left atrial volume index, as well as performing the statistical analysis with it.
Minor issues:
- Line 229: “However, we tried to maintain the quality of the voltage by excluding voltage points less than 800 points.” This sentence should be rephrased, the authors mean probably that voltage maps with less than 800 points were excluded (as elsewhere already stated: lines 64-65).
Best regards,
Author Response
We thank the editor and reviewers for their helpful and insightful comments. We believe that our revised manuscript, based on their comments and suggestions, has been markedly improved relative to its accuracy, readability, and the soundness of our conclusions. Please see the attatchment.

Round 2
Reviewer 3 Report
Dear Sir/Madam,
Thank you for reviewing the manuscript and addressing the mentioned issues. These were adequately answered. Therefore, the manuscript seems suitable for publishing in the present form.
Best regards